# Parental Involvement and Children’s Subjective Well-Being: Mediating Roles of the Sense of Security and Autonomous Motivation in Chinese Primary School Students

**DOI:** 10.3390/bs14070603

**Published:** 2024-07-16

**Authors:** Xiaoxian Liu, Hengyuan Fan, Xinya Shang, Wange Li, Xinhui He, Purui Cao, Xiaosheng Ding

**Affiliations:** Faculty of Education, Henan Normal University, Xinxiang 453000, China; 111028@htu.edu.cn (X.L.); 2310283157@stu.htu.edu.cn (H.F.); 2010283116@stu.htu.edu.cn (X.S.); 2210283146@stu.htu.edu.cn (W.L.); 2210183026@stu.htu.edu.cn (X.H.); 2110183026@stu.htu.edu.cn (P.C.)

**Keywords:** parental involvement, sense of security, autonomous motivation, well-being

## Abstract

Parental involvement may influence the subjective well-being of primary school students, but the specific mechanisms remain unclear. This study explores the mechanisms between parental involvement and primary school students’ subjective well-being. The current study investigated 340 fifth and sixth grade students and their parents from a primary school using the Parental Education Involvement Behavior Scale, the Well-Being Scale, the Sense of Security Scale, and the Learning Self-Regulation Scale. We found that (1) sense of security plays a complete mediating role between parental involvement and primary school students’ subjective well-being; (2) autonomous motivation has a marginal mediating effect between parental involvement and subjective well-being of primary school students; and (3) sense of security and autonomous motivation play a chain mediating role between parents’ educational involvement and primary school students’ well-being. In conclusion, parental involvement appears to contribute to primary school children’s subjective well-being, and this effect may be mediated individually and sequentially by the children’s sense of security and autonomous motivation.

## 1. Introduction

Numerous studies have underscored the crucial role of family education in enhancing children’s academic achievement, mental health development, and overall happiness [1,2,3]. The challenge of promoting high-quality and balanced family education development has become a significant concern [4]. Parental involvement, among other familial factors, is one of the most frequently studied variables in this context [5]. Researchers in psychology and education are interested in how parents can participate effectively in children’s education. In China, academic achievement is highly esteemed and viewed as a critical avenue for success and social mobility [6]. Research indicates that Chinese parents tend to be more actively involved in their children’s education from an early stage compared to their Western counterparts [7].

Parental involvement is conceptualized as a variety of parent practices related to children’s academic lives and social-emotional well-being at home and at school, including parents’ educational beliefs, academic achievement expectations, discussing school with children, and monitoring their progress [8,9]. Research has consistently shown that parental involvement significantly influences children’s academic performance, learning motivation, and subjective well-being [10,11,12,13,14,15]. Furthermore, Ryan and Deci [16] suggest that parental involvement can enhance children’s mental health by meeting their basic needs, including those for safety and autonomy. Consequently, we examined the relationship between parental involvement and children’s subjective well-being in the present study. We also explored the chain mediating roles of children’s sense of security and autonomous motivation within this relationship, aiming to contribute to the improvement of family education quality and children’s subjective well-being.

### 1.1. Parental Involvement and Children’s Subjective Well-Being

The term subjective well-being (SWB) refers to individuals’ self-assessed well-being across all aspects of their lives, encompassing life satisfaction, high positive affect, and low negative affect [17,18]. Bronfenbrenner’s Bioecological Model of Human Development, along with Developmental Contextualism, posits that the family acts as a proximal context, directly shaping a child’s development [19]. This influence occurs through continuous interactions within the “individual–family” microsystem, including positive parenting practices such as regulation, autonomy support, and parental involvement [20,21,22]. Numerous studies have confirmed that family environments and positive parenting practices significantly influence children’s SWB [23,24,25,26].

As mentioned above, parental involvement, an important practice of parenting, is associated with children’s SWB [27]. For instance, Yap and Baharudin [28] found that parental involvement was positively associated with adolescent positive affect and life satisfaction among adolescents in Malaysia. Similarly, Lv et al. [29] reported that parental involvement in parent–school communication enhanced children’s SWB in China. Moreover, a recent longitudinal study found that adolescents whose parents had more academic involvement were more likely to experience an increase in subjective well-being [30].

Based on the existing literature, we inferred that parental involvement may positively predict primary school students’ subjective well-being.

### 1.2. The Mediating Effect of Sense of Security between Parental Involvement and SWB

Although several studies have shown that parental involvement is associated with child and adolescent SWB, limited research has explored the factors that might explain this association. In this study, we examined two potential mediators: the child’s sense of security and autonomous motivation.

Maslow [31] defined psychological security as the sense of being liked or loved, belonging, safety, and calm, along with the absence of threat, conflict, or distress. Numerous studies have established that children’s basic need for psychological security is met through interactions with their parents. For instance, Attachment Theory [32] posits that the interaction between parent and child is crucial in fostering the child’s sense of psychological security. Additionally, Cong and An [33] identified that key components of psychological security, such as a sense of certainty and control, are provided by parents. Furthermore, research has consistently shown that an authoritative parenting style, a positive parent–child relationship, and significant paternal co-parenting positively predict the psychological security of children and adolescents [8,34,35,36]. 

Similarly, extensive evidence suggests a relationship between children’s sense of security and their SWB. Firstly, Maslow’s Hierarchy of Human Needs Theory posits that the absence of safety and security needs induces anxiety [37]. Secondly, Tay and Diener [38] argued that the fulfillment of basic needs, such as safety and security, correlates strongly with positive life evaluations and fewer negative feelings. Finally, Lee and Yoo [39] observed that children who perceive their family, school, and neighborhood environments as safe report higher subjective well-being. 

Based on these findings, we inferred that parental involvement may enhance primary school students’ SWB by increasing their sense of security.

### 1.3. The Mediating Effect of Autonomous Motivation between Parental Involvement and SWB

Autonomous motivation is defined as the inclination to organize behavior by aligning with personal interests, values, and the support received in interpersonal contexts [16]. Self-determination theory (SDT) posits that nutritive parenting practices, which include autonomy support, structure, and parental involvement, positively affect the development of intrinsic motivation [16]. Autonomous motivation encompasses identified, integrated, and intrinsic regulation, and as an important form of intrinsic motivation [40], it may also be positively influenced by nutritive parenting practices. Indeed, several studies have demonstrated that parental involvement has positive influences on children’s autonomous motivation [27,41]. 

Meanwhile, SDT posits that meeting the need for autonomy is essential for initiating and regulating behavior, and perceiving one’s actions as volitional contributes to enhanced subjective well-being [42]. For instance, a study in China demonstrated that the satisfaction of basic needs, including autonomy, significantly and positively predicts adolescents’ positive emotions and life satisfaction [43]. Another study revealed that adolescents who perceived greater autonomous support from their parents tended to be more adaptive, more engaged, and happier in their school life [8].

Thus, we inferred that autonomous motivation may mediate the relationship between parental involvement and primary school students’ subjective well-being.

### 1.4. The Chain Mediating Effect of Sense of Security and Autonomous Motivation

In the present study, we also examined the chain mediating roles of sense of security and autonomous motivation in the relationship between parental involvement and SWB among primary school students. Several well-established theoretical frameworks robustly support the hypothesis of a positive correlation between these variables.

Firstly, SDT suggests that the satisfaction of basic psychological needs, including autonomy, relatedness, and ability, is critical for developing autonomous motivation [16]. Numerous studies have shown that the fulfillment of these psychological needs can positively predict children’s autonomous motivation [44,45,46]. Secondly, according to Ryan and Deci’s Basic Psychological Needs Theory (BPNT) [47], a sub-theory of SDT, a sense of security serves as an essential indicator of the fulfillment of basic psychological needs. In environments characterized by safety and security, children are more likely to exhibit autonomous behaviors [47]. Similarly, Bronfenbrenner’s Ecological Systems Theory [48] posits that within supportive and secure microsystems, such as the secure family, children are more likely to demonstrate autonomous behaviors. Moreover, Attachment Theory posits that secure attachment relationships provide a stable base for children, enabling them to explore their environment and exhibit more autonomy in their behaviors [32].

Based on these theories and research, we hypothesized that sense of security and autonomous motivation serve as chained mediators in the relationship between parental involvement and subjective well-being among primary school students.

### 1.5. The Current Study

In the current study, we explored whether and how parental involvement predicts primary school students’ subjective well-being. The first objective was to examine the direct relationship between parental involvement and children’s subjective well-being. The second was to examine the individual and sequential mediating roles of psychological security and autonomous motivation in this relationship. The conceptual model is presented in Figure 1. We tested the following hypotheses:

**Hypothesis** **1:**
*Parental involvement will be directly and positively related to primary school students’ subjective well-being.*


**Hypothesis** **2:**
*Psychological security will mediate the association between parental involvement and primary school students’ subjective well-being.*


**Hypothesis** **3:**
*Autonomous motivation will mediate the association between parental involvement and primary school students’ subjective well-being.*


**Hypothesis** **4:**
*Parental involvement will predict subjective well-being through the chain mediating effects of primary school students’ sense of security and autonomous motivation.*


## 2. Materials and Methods

### 2.1. Participants

Families with students from three classes in both grades 5 and 6 at a primary school in Henan Province (a central region of China) were randomly selected using cluster sampling methods. A total of 388 students, along with their parents, participated in the research. After excluding invalid questionnaires, data from 340 students and their parents were eligible for analysis, indicating a response validity rate of 87.63%.

The student participants consisted of 188 5th graders (55.29%) and 152 6th graders (44.71%), with a gender distribution of 163 males (47.94%) and 177 females (52.06%). The average age of these students was 11.37 years (SD = 0.53). Among the parent participants, there were 112 fathers (32.94%) and 228 mothers (67.06%), with an average age of 37.97 years (SD = 4.23).

### 2.2. Procedure

This research was approved by the Ethics Committee for Psychological Research of the corresponding author’s institution. Before data collection, all participants (including parents and their children) gave written informed consent, indicating their understanding of the study processes and their agreement to participate.

Master’s students in psychology from our university and the head teachers of the classes from which the participants were drawn were recruited as research assistants to collect data. The recruitment process was internal, selecting candidates based on their academic qualifications and previous experience with similar research tasks. Prior to conducting the assessments, they received comprehensive training, which included the testing procedures, specific requirements, and relevant precautions.

The assessment was conducted in two ways: online and through on-site surveys. Under the guidance of the Master’s students, primary students completed self-reported questionnaires in their classrooms. The questionnaires addressed topics in the following sequence: subjective well-being, sense of security, and autonomous motivation. It took the students approximately 15 min to complete the questionnaires. They were instructed to respond based on their actual situations and reassured that there were no right or wrong answers. Additionally, with the assistance of the head teachers, the students’ parents completed an online survey on parental involvement, which was distributed through the Sojump platform.

### 2.3. Measures

#### 2.3.1. Parental Involvement

The Parental Educational Involvement Behavior Questionnaire for Primary School Students (Parent Version) is a Chinese-language measure of the extent to which parents of primary school students engage in positive behaviors relevant to their child’s education [40]. The questionnaire contains 29 items, divided into five dimensions: family monitoring (4 items, e.g., “I ask my children to get up at the same time in the morning and go to bed at the same time in the evening”), academic counseling (5 items, e.g., “I will help my children solve their difficulties in study”), parent–child communication (6 items, e.g., “I share stories from my school days with my children”), joint activities (7 items, e.g., “I take my children to some charity activities”), and home–school communication (7 items, e.g., “I talk to the teacher about the children doing their homework at home”). A 4-point rating system is used, ranging from 1 (never) to 4 (often), with higher scores indicating higher positive parental involvement. In this study, the internal consistency reliabilities of the dimensions ranged from 0.75 to 0.82, and the overall internal consistency reliability of the questionnaire was 0.90.

#### 2.3.2. Children’s Subjective Well-Being

The Flourishing Scale [49] was utilized to measure students’ self-reported subjective well-being. We used the Chinese version of the scale in the current study, which demonstrated good reliability and validity within the Chinese population [50]. It comprises 8 items, such as “I am optimistic about my future”, employing a 7-point scoring system, ranging from 1 (strongly disagree) to 7 (strongly agree), with higher scores indicating higher subjective well-being. In this study, the internal consistency reliability of the scale was 0.91.

#### 2.3.3. Children’s Sense of Security

The Sense of Security Scale [32] is a Chinese-language, self-reported measure of psychological security. The scale was developed using qualitative and quantitative methods in a Chinese sample. The scale consists of 16 items, divided into two dimensions: interpersonal security (8 items, reflecting an individual’s sense of security during interpersonal interactions, e.g., “I’m used to giving up my wishes and demands”); and certainty in control (8 items, reflecting an individual’s certainty of control; e.g., “I’m always worried that something bad will happen”, reverse-scored). The scale employs a 5-point rating system, ranging from 1 (strongly agree) to 5 (strongly disagree), with higher scores indicating a higher sense of security. In this study, the internal consistency reliabilities of the dimensions of interpersonal security and certainty in control were 0.84 and 0.89, respectively, with the overall scale having an internal consistency reliability of 0.92.

#### 2.3.4. Children’s Autonomous Motivation for Learning

The Learning Self-Regulation Questionnaire (SRQ-A) was developed as an English- language measure for students to self-report on their reasons for learning [51]. In the current study, we used the Chinese version of the questionnaire [52]. The total scores for the identified regulation (7 items, e.g., “because I would like to learn new knowledge”) and intrinsic motivation (7 items, e.g., “because it is very interesting for me to do homework”) dimension are used as indicators of autonomous motivation for learning. A 4-point scoring system is used, ranging from 1 (strongly disagree) to 4 (strongly agree), with higher scores indicating higher autonomous motivation. In this study, the internal consistency reliabilities of the four sub-dimensions ranged from 0.78 to 0.85, and the overall internal consistency reliability of the questionnaire was 0.92.

### 2.4. Data Analysis

SPSS 21.0 was used to test common method bias, generate descriptive statistics, and calculate correlations. Mplus 8.3 was used to test mediation.

## 3. Results

### 3.1. Test of Common Method Bias

Anonymity and reverse scoring were used to mitigate potential common method bias. The Harman single-factor test was utilized to evaluate the extent of common method bias [53]. The result revealed 19 factors with eigenvalues exceeding 1. The primary factor accounted for only 19.41% of the total variance, which is significantly below the critical threshold of 40%. This indicates the absence of a severe common method bias in this study.

### 3.2. Descriptive Statistics and Correlations

Descriptive statistics and correlations among variables are presented in Table 1. Significant positive correlations were found among the total scores for parental involvement, sense of security, autonomous motivation, and subjective well-being (*p* < 0.01). The dimension scores from each measure were also intercorrelated (*p* < 0.05), with two exceptions. The academic counseling dimension of parental educational involvement did not significantly correlate with the identified regulation dimension of autonomous motivation. Similarly, the home–school communication dimension did not significantly correlate with either the autonomous motivation total score or dimension scores.

### 3.3. Chain Mediating Effects of Sense of Security and Autonomous Motivation

A sequential mediation model was tested in which parental involvement was the predictor variable and pupils’ subjective well-being was the outcome variable, with sense of security and autonomous motivation serving as individual and sequential mediators. A test of the mediation effects was conducted with 1000 bootstrap samples. The results are shown in Figure 2 and Table 2.

The results indicated that the model exhibited an excellent overall fit: χ^2^/df = 2.746, CFI = 0.966, TLI = 0.948, RMSEA = 0.069. Preliminary analyses of the associations among variables were consistent with possible mediation effects. Parental involvement did not significantly predict subjective well-being directly (β = 0.033, *p* > 0.05), but it was significantly and positively correlated with sense of security as a potential mediator (β = 0.272, *p* < 0.001) and marginally significantly and positively correlated with autonomous motivation as a potential mediator (β = 0.142, *p* = 0.052). Furthermore, sense of security was a significant positive predictor of both autonomous motivation (β = 0.421, *p* < 0.001) and subjective well-being (β = 0.438, *p* < 0.001), while autonomous motivation also significantly and positively predicted subjective well-being (β = 0.425, *p* < 0.001).

There was a significant mediation effect of sense of security in the association between parental involvement and subjective well-being (effect = 0.119, *p* < 0.001), with a 95% confidence interval of [0.068, 0.178]. There was also a marginally significant mediation effect of autonomous motivation in the association between parental involvement and subjective well-being (effect = 0.060, *p* = 0.051), with a 95% confidence interval of [−0.001, 0.121]. The chain mediating effect of sense of security and autonomous motivation was significant (effect = 0.049, *p* < 0.001), with a 95% confidence interval of [0.027, 0.085].

## 4. Discussion

The current study examined the relationship between parental involvement and primary school students’ subjective well-being, and the chain mediating effect of the child’s sense of security and autonomous motivation in this association. Parental involvement significantly predicted children’s subjective well-being indirectly through the mediating effect of the child’s security as well as through the chain mediating effect of security and autonomous motivation. These results partially supported the hypothesized model.

Parental involvement indirectly and positively predicted the subjective well-being of primary school students. Specifically, the more parents participated in their children’s educational and learning activities, the higher the subjective well-being that the children experience, which is consistent with previous research [8]. For primary school students, parental involvement in counseling, parent–child communication, and joint activities could enhance their learning abilities, guide them in appropriately addressing various problems encountered at school, and alleviate negative emotions, which allows them to experience more care and naturally improves their sense of well-being [54,55].

The analysis of mediating effects indicated that parental involvement predicts primary students’ subjective well-being through enhancing the child’s sense of security. Parental involvement provides encouragement and support for children, playing a crucial role in developing their sense of security [11]. While parental involvement may sometimes entail negative behaviors like psychological and behavioral control, the boundaries set by parents contribute to a stable sense of belonging and security [56]. Notably, compared to Western children, Chinese pupils are more inclined to view parental control and educational supervision as expressions of care and love [57]. Consequently, greater parental involvement in China may lead to pupils’ higher security. Additionally, according to Maslow’s Hierarchy of Needs theory, well-being represents the ultimate expression of security. Thus, when individuals maintain a high level of security, they also experience a high level of well-being [58,59]. In summary, parental involvement could indirectly enhance children’s subjective well-being by fostering their sense of security.

The association between parental involvement and children’s autonomous motivation was marginally significant, and the mediating effect of autonomous motivation in the association between parental involvement and subjective well-being was also marginally significant. Although some studies have found that parental involvement significantly predicted primary school students’ autonomous motivation [10,41,60], self-determination theory suggests that parental involvement, autonomy support, and structuring play distinct roles in satisfying children’s basic needs. Specifically, parental autonomy support fulfills children’s need for autonomy, contributing to the development of their autonomous motivation. In contrast, parental involvement primarily satisfies children’s needs for relatedness and competence, more effectively fostering their sense of security [16]. Thus, some researchers suggest that parental involvement, which accompanied by autonomy support, is more important to developing children’s autonomous motivation [27]. For instance, a recent longitudinal study found that adolescents who perceived greater autonomous support from their parents tended to be more adaptive, more engaged, and happier in their school life [8].

Studies in China have found that most parents have high educational expectations for their children and often monitor their lives and learning, with limited autonomy support, leading to a relatively low fulfillment of children’s autonomy needs compared to other cultural backgrounds [16,61]. Satisfying the child’s autonomy needs is crucial for developing autonomous motivation, and this might explain why there was only a marginally significant association between parental involvement and elementary school students’ autonomous motivation [42]. In addition, pupils’ autonomous motivation significantly predicted subjective well-being in the current study, consistent with previous research [8,11,59]. According to self-determination theory, when children autonomously make choices within the constraints of their environment, their basic psychological needs are met, fostering internal motivation [42,45]. This fulfillment brings significant psychological satisfaction to children, thus enhancing their subjective well-being [8,62].

Finally, our results suggest that parental involvement might positively influence primary school students’ subjective well-being through the chain mediating effects of the child’s sense of security and autonomous motivation. Self-determination theory maintains that a sense of security is a necessary condition for healthy development and is an important source of motivation [16,63]. The theory also proposes that the satisfaction of children’s relational needs can promote a sense of security that enables the development of intrinsic motivation, the highest form of autonomous motivation [16]. Therefore, higher parental involvement could lead to the child’s greater sense of security, greater autonomous motivation, and ultimately higher subjective well-being.

## 5. Limitations and Further Research

Although the current study offers several important insights, its limitations should also be considered and may provide directions for future research.

First, there might have been sample selection bias, as the participants came from only one primary school in China and there was a limited sample size. In other words, the relatively singular psychological characteristics of children in the current study may restrict the generalizability of our findings. It is not clear whether our findings can be extended to pupils living in other countries and regions with more diverse social characteristics and family educational models. Thus, more research with larger and more diverse samples is needed to verify the current findings.

Second, given that all data in the current study were collected at the same time point, the cross-sectional design cannot demonstrate causal relationships. Future research should employ both experimental designs (e.g., interventions) and longitudinal designs (e.g., cross-lagged studies) to confirm the causal relations between parental involvement and subjective well-being.

Third, only self-report measures were used, and the limitations associated with this type of assessment should be considered. Specifically, the observed relations in the current analyses may be influenced by subjective bias. It is recommended that future research should use multi-method (e.g., behavioral) and multi-informant (e.g., teachers and peers) measures to replicate our findings.

Finally, another key factor to consider is the cultural context in which parental involvement was defined and measured. As mentioned above, Chinese pupils are more inclined to interpret parental control and supervision in educational involvement as expressions of care and love [57], which implies that high parental involvement in children’s education may lead to a high sense of security for children in China. However, this belief may not be shared by other cultures that have different norms and values regarding parenting. Therefore, we recommend that researchers attempt to replicate the current findings in different ethnic and cultural groups as a way to establish the external validity of the results.

## 6. Practical Implications

The present study has important implications for educators and parents. Firstly, the study emphasizes the crucial role of parental involvement in enhancing subjective well-being among primary school students, particularly by strengthening their sense of security and autonomous motivation. Secondly, based on these findings, educators can develop school policies and activities that promote parental involvement, such as parent–school interaction programs and homework support sessions. Moreover, teachers and school administrators should advocate for active parental roles in the educational process by providing clear communication channels and supportive strategies to facilitate effective parental participation in their children’s learning and development.

## 7. Conclusions

Parental involvement appears to contribute to primary school children’s subjective well-being, and this effect may be mediated individually and sequentially by the children’s sense of security and autonomous motivation. On one hand, parental involvement may promote both a sense of security and autonomous motivation, and each of these could increase children’s subjective well-being, leading to increased feelings of well-being. On the other hand, the association between parental involvement and the child’s sense of security could in turn create greater autonomous motivation, resulting in higher subjective well-being.

## Figures and Tables

**Figure 1 behavsci-14-00603-f001:**
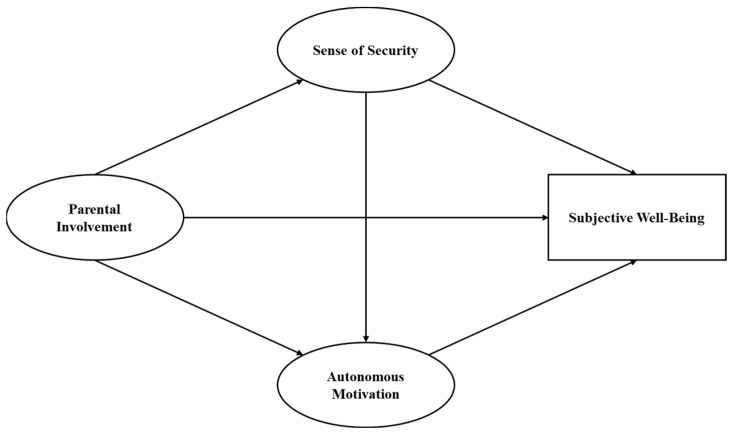
Hypothesized model: Parental involvement is expected to be associated with primary school children’s subjective well-being through the individual and chain mediating effects of sense of security and autonomous motivation.

**Figure 2 behavsci-14-00603-f002:**
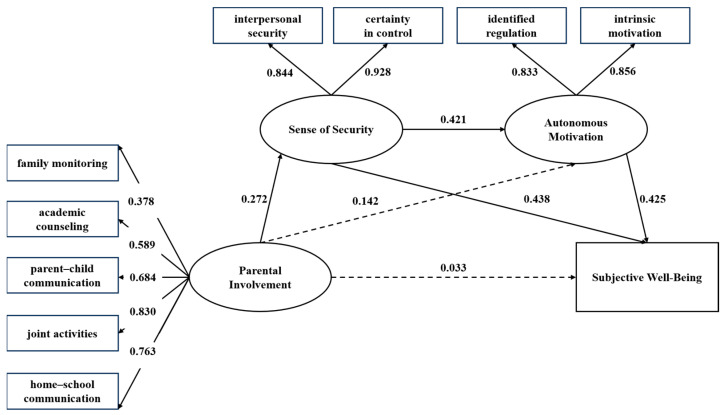
Chain mediating effects of sense of security and autonomous motivation in the association between parental involvement and children’s subjective well-being.

**Table 1 behavsci-14-00603-t001:** The correlation analysis of main variables.

	*M ± SD*	1	2	3	4	5	6	7	8	9	10	11	12
1 sense of security	3.78 ± 0.93	1											
2 interpersonal security	3.92 ± 0.88	0.93 **	1										
3 certainty in control	3.65 ± 1.08	0.96 **	0.78 **	1									
4 parental involvement	2.95 ± 0.44	0.25 **	0.27 **	0.21 **	1								
5 family monitoring	3.25 ± 0.54	0.15 **	0.14 **	0.13 *	0.48 **	1							
6 academic counseling	3.15 ± 0.65	0.21 **	0.24 **	0.17 **	0.70 **	0.30 **	1						
7 parent–child communication	3.37 ± 0.52	0.21 **	0.24 **	0.17 **	0.77 **	0.32 **	0.51 **	1					
8 joint activities	2.69 ± 0.61	0.20 **	0.20 **	0.18 **	0.84 **	0.21 **	0.44 **	0.56 **	1				
9 home–school communication	2.56 ± 0.61	0.16 **	0.17 **	0.13 *	0.80 **	0.26 **	0.37 **	0.44 **	0.66 **	1			
10 identified regulation	3.19 ± 0.67	0.34 **	0.33 **	0.32 **	0.17 **	0.15 **	0.09	0.17 **	0.15 **	0.09	1		
11 intrinsic motivation	2.63 ± 0.81	0.41 **	0.38 **	0.40 **	0.17 **	0.11 *	0.11 *	0.17 **	0.16 **	0.09	0.72 **	1	
12 autonomous motivation	2.91 ± 0.69	0.41 **	0.38 **	0.39 **	0.19 **	0.14 *	0.11 *	0.18 **	0.17 **	0.10	0.91 **	0.94 **	1
13 subjective well-being	5.43 ± 1.31	0.62 **	0.55 **	0.61 **	0.22 **	0.11 *	0.12 *	0.17 **	0.20 **	0.17 **	0.53 **	0.52 **	0.57 **

Note: * means *p* < 0.05; ** means *p* < 0.01.

**Table 2 behavsci-14-00603-t002:** The results of the mediating effect analysis.

Models	Paths	β	95% CI	Relative Effect (%)
Direct effect	PI→SWB	0.033	[−0.059, 0.120]	12.69
Indirect effect	PI→SFS→SWB	0.119	[0.068, 0.178]	45.77
	PI→AM→SWB	0.060	[−0.001, 0.121]	23.08
	PI→SFS→AM→SWB	0.049	[0.027, 0.085]	18.46
Total effect	-	0.26	-	-

Note: PI = Parental Involvement; SFS = Sense of Security; AM = Autonomous Motivation; SWB = Subjective Well-Being. Standardized regression coefficients are shown.

## Data Availability

No new data were created or analyzed in this study. Data sharing is not applicable to this article.

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
