# Peer review of "Parental Involvement and Children’s Subjective Well-Being: Mediating Roles of the Sense of Security and Autonomous Motivation in Chinese Primary School Students"

_behavsci, 2024, doi:10.3390/bs14070603_

Round 1

Reviewer 1 Report

Comments and Suggestions for Authors

Thank you for the opportunity to review your article on parental involvement in the subjective well being of children. I found your study intriguing as the concepts/variables you tested were subjective and based on self reports, thus I wondered how these would be measured and conclusions made.  I have made comments on your script so please refer to these. 

Overall, I found from the Results and Discussion sections to be well structured and well written. I enjoyed reading these as they flowed and were clear and succinct. Limitations were accurately identified and conclusions were sound (See comment on implications below).  

In contrast, I found the Introduction and Methods not well structured nor well written. Likewise, there are issues with the Methods section because, as written, this study could not be replicated - details are needed. The first paragraph in the Introduction appears random with too many concepts added together but without any order or explanation. The context needs to be explained clearly. Likewise, you should consider placing the two theoretical frameworks together and then justify your use of Maslow's theory. As written these are separate so it is difficult to follow their importance/relevance. Consider also refining/structuring the definitions section of SWB - pull out the main features. As written, this section appears slightly random and is difficult to follow  - structure carefully.  

Methods: clarity/detail is needed in this section.  Refer to my comments on the script. This section must be replicable, as written it is not. 

Discussion / conclusion. There is mention in the Conclusions about the implications for educators from the findings of this study. These implications need to be placed in the Discussion section and explained, otherwise the sentence in the Conclusion needs to be removed (do not add anything new to the conclusions).   

Overall, there is much potential in your article and with the front sections revised, this will make a much stronger article for publication. 

Comments on the Quality of English Language

Please check your sentence construction in your Introduction and Methods sections. On occasions, this is jumbled, thus difficult to follow.  

Author Response

Dear reviewer:

Thank you for your comments concerning our manuscript entitled “Parental Involvement and Children’s Subjective Well-Being: Mediating Roles of the Sense of Security and Autonomous Motivation in Chinese Primary School Students” (behavsci-3052524). These comments are very valuable and helpful for the revisions and improvement of our manuscript. We have made revised according to the suggestion.

Please see the attachment: 

Reviewer 2 Report

Comments and Suggestions for Authors

It was very interesting to read your research paper, thank you very much.

My attention was drawn to the fact that parental involvement was mentioned, but not explicitly defined. In order to make your study more clear, could you please provide an explanation for what parental involvement is and how it is operationalized? It would be helpful to readers if you included a description of how you assessed or conceptualized this very significant phenomenon (operationalization is in the text).

Please do not use such simple statements in your introduction. Rather, I would like you to place the study in a broad context and highlight why the study is necessary.

In the study, the Bioecological Model of Human Development and Developmental Contextualism falls into the category of Urie Bonfenbrenner's model. However, Bonfenbrenner's name is not mentioned in the study. I do believe there is potential for it to be successful, but it lacks an adequate foundation on which to build, although I believe it has some potential.

According to your introduction, we do not know what effect PI has on SWB, and we do not know how it affects SWB. We do not have complete knowledge of it, but we do know more than what is mentioned here. The author, for example, can shed more light on the topic if he or she includes the following literature in the paper.

Treviño, Ernesto, et al. "Socioeconomic status, parental involvement and implications for subjective well-being during the global pandemic of Covid-19." Frontiers in Education. Vol. 6. Frontiers Media SA, 2021.

Li, Ruoxuan, et al. "Chinese parental involvement and adolescent learning motivation and subjective well-being: More is not always better." Journal of Happiness Studies 21 (2020): 2527-2555.

Khan, Aqeel, et al. "Does psychological strengths and subjective well-being predicting parental involvement and problem solving among Malaysian and Indian students?." SpringerPlus 3 (2014): 1-6.

Li, Simeng, et al. "The Developmental Trajectory of Subjective Well-Being in Chinese Early Adolescents: The Role of Gender and Parental Involvement." Child Indicators Research (2024): 1-22.

The methodology is relevant to the study. When you are using the SPSS, it may be more obvious to use the Amos version instead of the Mplus version if you are using the SPSS. In spite of that, it is not a gap at all. There are adequate fit indicators for the model. I would like to congratulate you on the high response rate you received

It is a major shortcoming and must be remedied: The study does not include any descriptive statistics so it cannot be used as a descriptive study.

In this discussion, you don't go into enough detail, you could use the new literature to enhance the discussion.

Author Response

Dear reviewer:

Thank you for your comments concerning our manuscript entitled “Parental Involvement and Children’s Subjective Well-Being: Mediating Roles of the Sense of Security and Autonomous Motivation in Chinese Primary School Students” (behavsci-3052524). These comments are very valuable and helpful for the revisions and improvement of our manuscript. We have made revised according to the suggestion. Revised portions are marked in red in the manuscript. The main revision in the manuscript and the responses to the reviewers’ comments are as follows:

Please see the attachment:

Round 2

Reviewer 2 Report

Comments and Suggestions for Authors

Thank you for the corrections. I accept it.